# CRPC Membrane-Camouflaged, Biomimetic Nanosystem for Overcoming Castration-Resistant Prostate Cancer by Cellular Vehicle-Aided Tumor Targeting

**DOI:** 10.3390/ijms23073623

**Published:** 2022-03-26

**Authors:** Kai Lu, Zheng Li, Qiang Hu, Jianfei Sun, Ming Chen

**Affiliations:** 1Institute of Urology, Southeast University Medical College, Nanjing 210009, China; kai.5391@hotmail.com (K.L.); 230189858@seu.edu.cn (Q.H.); 2The State Key Laboratory of Bioelectronics and Jiangsu Key Laboratory of Biomaterials and Devices, School of Biological Sciences and Medical Engineering, Southeast University, Nanjing 210009, China; lizssy18120131686@outlook.com

**Keywords:** oncotherapy, homotypic target, drug delivery system, nanoparticles, docetaxel

## Abstract

Castration-resistant prostate cancer (CRPC) is the most common malignant tumor of the male urinary system. Nanodrug delivery systems (NDDS) have been widely applied in drug delivery for tumor therapy; however, nanotherapeutics encounter various biological barriers that prevent successful accumulation of drugs, specifically at diseased sites. Therefore, there is an urgent need to develop a CRPC-targeting nanocomposite with fine biocompatibility for penetrating various biological barriers, delivering sufficient drugs to the targeting site and improving therapeutic efficiency. In this work, CRPC cell membranes were firstly adapted as biomimetic vectors for the encapsulating PEG−PLGA polymer containing the chemotherapy drug docetaxel (DTX). The CRPC membrane-camouflaged nanoparticles can easily escape early recognition by the immune system, penetrate the extracellular barrier, and evade clearance by the circulatory system. In addition to the characteristics of traditional nanoparticles, the CRPC cell membrane contains an arsenal of highly specific homotypic moieties that can be used to recognize the same cancer cell types and increase the targeted drug delivery of DTX. In vivo fluorescence and radionuclide dual-model imaging were fulfilled by decorating the biomimetic nanosystem with near-infrared dye and isotope, which validated the homotypic targeting property offered by the CRPC cell membrane coating. Importantly, remarkably improved therapeutic efficacy was achieved in a mice model bearing CRPC tumors. This homologous cell membrane enabled an efficient drug delivery strategy and enlightened a new pathway for the clinical application of tumor chemotherapy drugs in the future.

## 1. Introduction

Prostate cancer is the most common malignant tumor of the male urinary system, and the incidence of prostate cancer in European and American countries is high, second only to lung cancer [1]. Although the early stages of prostate cancer can be treated with surgery, radiotherapy, androgen deprivation therapy (ADT), etc., most patients with advanced prostate cancer eventually develop CRPC [2] and have intrinsic or acquired resistance to ADT and other hormone therapy [3]. Many drugs (abiraterone acetate [4], cabozantinib [5], docetaxel [6], etc.) have been developed for the treatment of CRPC. Among them, docetaxel is the first cytotoxic chemotherapy drug approved by the US Food and Drug Administration (FDA) for CRPC [7]. However, traditional drug delivery methods have disadvantages of poor targeting and low drug utilization rate, which greatly reduces the therapeutic effect of the drug. In addition, drugs can also cause great damage to normal human cells, resulting in a variety of adverse reactions (hematological toxicity, allergic reactions, and immunosuppression) [8,9]. Therefore, the safe, efficient, and targeted delivery of drugs has a positive significance for improving drug utilization efficiency, reducing drug damage to the human body, and improving the therapeutic effect of CRPC.

In recent years, with the rapid development of oncology and nanomaterials, nanodrug delivery systems (NDDS) have shown remarkable therapeutic effects in tumor diagnosis and treatment, and a variety of nanodrug carriers have been applied to tumor treatment [10,11,12,13]. Compared with drug delivery methods, NDDS have the advantages of improving drug utilization, achieving stimulated release, and reducing drug toxicity and side effects [14,15,16,17,18]. Common nanocarriers in NDDS include liposomes, polymer micelles, polymer vesicles, dendrimers, hydrogels, and inorganic nanoparticles [19,20,21,22]. Among them, the hydrogel nano delivery system has the advantages of an excellent drug-carrying capacity, biocompatibility, degradability, and easy modification; it is also widely used in therapeutic drug delivery systems [23,24]. For example, Huang P et al. [25] prepared multifunctional polyelectrolyte chitosan nanoparticles (HA-g-PCL NPs), which entered cancer cells (EC109) through receptor-mediated endocytosis; normal fibroblasts (NIH3T3) were rarely absorbed, which overcame the shortcomings of the poor water solubility of lipophilic anticancer drugs and the disorders of the gastrointestinal microenvironment and systemic circulation. However, the actual application gel nanocarriers encounter multiple extracellular barriers, such as serum protein adsorption and reticular endothelial system (RES) clearance. In addition, owing to their exogenous nature, most nanoparticles are quickly cleared by the mononuclear phagocytic system. Therefore, there is an urgent need to develop a targeted nanocomposite for penetrating the cell barrier and deliver the drug to the target area to realize the targeted therapy of the drug and improve the therapeutic efficiency. To overcome these barriers, cell membrane-camouflaged nanoparticles have emerged as an innovative solution to solve this problem [26,27]. Cellular membranes used for camouflaging nanoparticles are usually isolated from immune cells, cancer cells, blood cells, fibroblast cells, and stem cells [28,29,30,31]. The excellent biocompatibility and versatile functionality of cell membranes have obviously increased their circulation and targeting efficacy, such as the widely studied blood cell membranes. However, the advantages of blood cell membranes were inclined to concentrate on a high drug loading efficiency, enhanced stability, and long circulation in body, which were regarded as natural advantages of cell membranes. To increase targeting distribution, blood cell membrane-coated nanoparticles generally required additional targeting support to navigate their tumor targeting applications. For example, a targeting peptide, such as RGD, was extensively applied to attach on to the surface of blood cell membranes and enhance their targeting ability [32,33], but this increased the workload of the preparation process. Compared with blood cell membranes, cancer cell membranes can achieve natural, superior tumor targeting through self-recognition via their surface-specific binding of membrane proteins. Based on the above studies, we selected an easily modified gel material (polylactic-glycolic acid (PLGA) NPs) with good biocompatibility and drug-carrying properties as a nanocarrier, and a new cancer cell membrane-camouflaged nanoparticle was prepared by modifying PLGA NPs containing DTX (PDNs) with the CRPC cell (DU145) membranes for targeted therapy of homotypic CRPC. Due to the homologous binding properties of the unique membrane proteins on the surfaces of cancer cell membranes, cell membrane-camouflaged nanoparticles exhibit strong homologous targeting abilities both in vivo and in vitro and can easily escape early recognition by the immune system, penetrate the extracellular barrier, and evade clearance by the circulatory system. The crucial proteins of DU145 cell membrane-encapsulated PDNs (CPDNs), including CD44, E-cadherin, CD147, and CD47, were identified by the western blotting analysis to confirm that proteins from the DU145 cell membranes were successfully retained in the CPDNs. The model of tumor cells (DU145 cells) were used to confirm CPDNs had a stronger binding efficiency to DU145 cells than RBC-camouflaged NPs and naked PDNs in vitro (Figure 1). To further evaluate the in vivo tumor selectivity, fluorescent and SPECT-CT imaging were used to assess the biodistribution of CPDNs, drug delivery, and therapeutic efficacy in mice. Our study describes the prospective development of a new method for personalized chemotherapy of CRPC using CPDNs, showing a favorable tumor-targeting ability and a potent tumor inhibition effect, which provides a new pathway for the clinical application of tumor chemotherapy drugs in the future.

## 2. Results and Discussion

### 2.1. Preparation and Characterization of DU145 CPDNs

We first synthesized PLGA-loaded docetaxel nanoparticles (PDNs) as the core, collected purified DU145 cell membranes (CMs) as the cancer cell biomimetic shell, and then employed CMs for the surface coating of PDNs. The morphology of NPs was observed by transmission electron microscopy (TEM). The morphology of CPDNs was a core−shell structure with a spherical shape, and the size of the CPDNs was 125.4 ± 0.6 nm, which was larger than that of PDNs (102.2 ± 0.5 nm) owing to the existence of CMs in CPDNs (Figure 2A). Dynamic light scattering (DLS) was used to measure the NP diameter and surface charge. The particle diameter was very similar to the outcomes measured by TEM, and the surface zeta potential of CPDNs was very close to the level of the membrane vesicles after being coated (Figure 2B,C).

Additionally, we identified crucial proteins in the CPDNs using western blotting, which revealed that the proteins from the DU145 cell membranes were successfully retained in the CPDNs. Characteristic signature proteins, including CD44, E-cadherin, CD147, and CD47, were identified by western blotting analysis. These membrane proteins are activating receptors found on prostate cancer cells that have crucial effects on the homotypic interactions and adherence abilities of tumor cells to metastasize to distant sites. Moreover, CD47 has been proven to prevent cancer cells from taking up macrophages and RES, enable NPs to escape immunogenic clearance [30], and provide NPs with long circulation times. Characteristic signature proteins, such as CD44, E-cadherin, CD147, and CD47, were identified by western blotting with the control Na/K ATPase (Figure 2D). The colocalization of CMs (red) and PDNs (green) internalized in DU145 cells was observed by scanning fluorescence microscopy images. A yellow color presented the successful colocalization of CM and PDN signals and indicated that CPDNs retained their structure particularly stably (Figure 2E).

Furthermore, the docetaxel release characteristics of CPDNs were investigated in the PBS and 1640 cell culture medium to measure the docetaxel stability in CPDNs, which is rightly related to toxicity. The DTX encapsulation rate was 52.5% with a DTX concentration of 0.1 mg/mL, and CPDNs barely encapsulated more DTX even with increasing concentrations of DTX (Appendix A). In Appendix A, the release percentage of docetaxel from CPDNs was found to be lower than that from PDNs in both the PBS and 1640 cell culture medium, which might be due to the outer shell membrane acting as a diffusion barrier for drug diffusion to release [34,35], indicating the notable stability of docetaxel in the fabricated CPDNs. To investigate the stability of PDNs and CPDNs in vitro, size variation was separately measured in the PBS and 1640 cell culture medium at different time points. As illustrated in Appendix A, CPDNs exhibited negligible changes in size in the PBS and 1640 cell culture medium whereas a slight increase in PDN size was observed in the 1640 cell culture medium, presumably caused by the combination of PDNs and proteins in 1640 cells. The cell membrane could enhance the stability of PDNs except for homotypic targeting competence.

### 2.2. Cellular Uptake of CPDNs

The model of tumor cells (DU145 cells) had an obviously higher uptake of CPDNs than PDNs and red blood cell membrane-coated PDNs (RPDNs) by fluorescence microscopy, indicating the specific ability of CPDNs to target homotypic cancer cells (Figure 3A). The cellular uptake of CPDNs was approximately 42 times and 20 times higher than those of RPDNs and PDNs, respectively, as measured by flow cytometric analysis to quantify the difference (Figure 3B). To validate the capability of DTX to be delivered into cells, CPDNs and PDNs labeled with ICG were incubated with DU145 and PrEC cells. After removing free CPDNs and PDNs, the cells were evaluated with flow cytometry (Figure 3C). DU145 cells exhibited an apparent increase in fluorescence intensity compared with a normal human prostate cell line, PrEC cells, which suggests that CPDNs could be specifically internalized by DU145 cells instead of PrEC cells. There was no obvious difference between PDNs and CPDNs in DU145 cells. The results demonstrated that the membrane coating could improve the targeting effect of CPDNs and advance their internalization by DU145 cells. All the results verified that the CPDNs were of superior selectivity to DU145 cells in vitro, indicating that CPDNs retain their active targeting ability owing to the transference of cells characteristic of crucial homotypic binding proteins. MTT assays were performed to assess the cytotoxicity of CPDN delivery to DU145 cells. Negligible cytotoxicity induced by different detected concentrations indicated the good biocompatibility of CPDNs (Figure 3D). Annexin V (V-FITC)-propidium (PI) double staining assays were performed to validate DU145 cell apoptosis by flow cytometry after treatment with CPDNs. As shown in Appendix A, the assay results demonstrated that CPDNs have a close relationship with cell apoptosis. The apoptosis ratio induced by CPDNs (13.5%) was obviously higher than that induced by PDNs (2.05%).

### 2.3. Biocompatibility Effect In Vivo

With the assurance of CPDN stability, biocompatibility, and in vitro cancer targeting, we next evaluated the distribution of CPDNs in a DU145-implanted tumor mouse model in vivo. The results revealed obvious differences in CPDN accumulation in the tumor from 0.5-, 1-, 2-, 4-, 12-, and 24-h in vivo tumor imaging (Figure 4A). The fluorescence intensity in tumors is positively correlated with the tumor-targeting ability of CPDNs. At the 24-h time point, the high accumulation of CPDNs in the tumor was still observed in vivo, indicating that CPDNs had a significantly long circulation time and notable homotypic binding ability. In the DU145-implanted tumor nude mouse model, after injecting PDNs/CPDNs into the mouse tail vein, we collected blood from the eye socket at different times. The fluorescence intensity measurement showed that the CPDNs remained for a longer time than the PDNs in the blood circulation. Furthermore, we dissected the mice to observe tissue metabolism using immunofluorescence, corresponding to fluorescence results (Figure 4B). Obvious fluorescence was still observed 24 h postinjection in the tumors of the CPDN group while negligible fluorescence was observed in the tumors of the PDN group. In addition to tumors, PDNs and CPDNs were also distributed in the kidney and liver. Finally, the advantage of CPDNs in tumors was demonstrated using the SPECT-CT system. Corresponding to bioimaging data in vivo, SPECT-CT more clearly showed that CPDNs could deeply penetrate prostate cancer, further confirming that the engineered CPDNs could homotypically target prostate cancer and have a long-term existence in the circulation (Figure 5A,B). Afterwards, the tissue distributions of CPDNs, PDNs, and DTX were determined in the tumor, kidney, liver, and plasma (Appendix A).

#### Treatment of CRPC

The ability of homotypic targeting of DU145 cells and chemotherapy effects provide CPDNs with therapeutic potency for treating CRPC in vivo. The model of DU145-implanted tumor mice was designed to evaluate the chemotherapy effect of CPDN. After one week of treatment, there was almost no difference in tumor growth in any of the groups. However, at subsequent times, tumor growth in the CPDN group was significantly inhibited compared with that in the docetaxel group and the PDN group. In addition, the antitumor growth of the PDN group was stronger than that of the docetaxel group, which was consistent with previous literature reports (Figure 6A) [36,37]. The median survival time of the CPDN group was longer than that of the other groups while the docetaxel and PLGA groups showed no significant difference (Figure 6B). Body weight is a crucial indicator for assessing the toxicity of chemotherapy. In our study, the weight loss of the CPDN group was significantly less than that of the other groups, suggesting that CPDNs have fewer toxic side effects and show better chemotherapy tolerance (Figure 6C). Finally, we performed immunohistochemistry to evaluate the expression levels of CD34 and Ki67 in prostate cancer after treatment [38,39]. The results showed that the protein expression levels of Ki-67 and CD34 in the CPDN group were decreased, indicating that CPDNs significantly inhibited the tumorigenicity and angiogenesis of the tumor compared with the other two groups, showing stronger anticancer properties (Figure 6D). Above all, the CPDNs could effectively inhibit the growth of CRPC in vivo.

## 3. Conclusions

At present, the main obstacles in the clinical application of traditional synthetic nanoparticles are early recognition by the immune system and rapid clearance by the circulatory system, eventually leading to low accumulation in the targeting organs. Recently, polymer NPs coated with cell membranes of target tissues were designed to mimic their biospecificity and overcome these obstacles [40,41]. In this study, a novel cancer cell membrane-coated biomimetic nanoparticle was prepared by encapsulating (PLGA) NPs containing DTX (PDNs) with CRPC cell (DU145) membranes, which was favorable to reduce the leakage of DTX to achieve targeted chemotherapy of CRPC. Through in vivo fluorescence and radionuclide dual-model imaging, CPDNs were validated by the homotypic targeting property offered by CRPC cell membrane coating. Importantly, a remarkably improved therapeutic efficacy was achieved in the mice model bearing CRPC tumors. In short, our design could increase the specific accumulation of chemotherapeutic drugs in the tumor site and prolong the retention time in blood circulation, thereby reducing the toxic side effects on the system and providing good chemotherapy tolerance. It is believed that CPDNs could provide new ideas for the specific treatment of CRPC. In the future, cell membrane-coated nanoparticles would be more widely used in tumor-specific therapy as an emerging drug delivery technology.

## 4. Experimental Section

### 4.1. Materials

PLGA (LA:GA = 50:50, Mn ≈ 25,000) was purchased from Jinan Daigang Biomaterial Co.,Ltd. (Jinan, China). The docetaxel, poly(allylamine hydrochloride) (PAH), and other chemicals were purchased from Sigma-Aldrich (St. Louis, MI, USA). Primary antibodies against CD44, E-cadherin, CD147, and CD47 were purchased from Santa Cruz Biotechnology. The secondary horseradish peroxidase (HRP)-linked, antimouse IgG antibody was procured from Cell Signaling. The fluorophore-conjugated antibody was purchased from BD Biosciences. The DU145 and PrRC cell lines were from the cell bank of the Shanghai Institute of Biological Sciences. The cell culture medium and fetal bovine serum were purchased from Invitrogen (Carlsbad, CA, USA).

### 4.2. Preparation of DU145 Cell Membranes

Similar to the method to synthesize MDINPs [42], 5 × 10^8^ DU145 cells were collected from the plate after centrifugation (1000 rcf, 5 min) at 4 °C. The resulting packed DU145 cells were resuspended in ice cold PBS. Then, we repeated the procedure two times, discarded the supernatant fluid, and added 4 mL of RIPA lysis buffer (medium, Beyotime, Shanghai, China). An ultrasonic processor (VCX150, sonic) was used to crush the mixture in ice water. The parameters were as follows: timer for 30 min, pulse on for 2 s, off for 3 s, and 30% amplification. The centrifuge was set at 12,000 rcf, 4 °C, for 4 min, and we collected the supernatant 3 times. The supernatant was centrifuged by Optima MAX-XP, Beckman Coulter ultracentrifugation (20,000 rcf, 4 °C, 30 min), the precipitate was discarded, and then further ultracentrifugation (100,000 rcf, 4 °C, 45 min) followed. The next steps were to discard the supernatant, dissolve the pellets in PBS, and store at −20 °C.

### 4.3. Preparation of PDNs

Docetaxel-loaded NPs were prepared from PLGA using a modified emulsification/evaporation method [43]. Briefly, the polymer and an appropriate amount of drug were dissolved in ethyl acetate to obtain a solution of 10% (*w/v*) PLGA and 1.5 mg/mL (*w/v*) docetaxel and then dissolved in 2.2% (*w/v*) polyvinyl alcohol (PVA) solution. Then, the mixture was vigorously shaken and ultrasonicated. The resulting nanosuspension was magnetically stirred overnight to allow the organic solvent to evaporate. Continuous ultracentrifugation/washing with distilled water can obtain the purification of NPs. Finally, the separated NPs were freeze-dried into fine cakes and stored at −20 °C for further use.

### 4.4. CPDN Synthesis

CPDNs were prepared using a previously reported extrusion approach [44]. In order to prepare cancer cell membrane vesicles, the above-mentioned membrane material was physically squeezed 11 times through a 400 nm polycarbonate membrane. Then, the vesicles and the core were coextruded through a 200 nm polycarbonate film, and the formed vesicles were coated on the PDNs.

### 4.5. Characterization of CPDNs

The size and surface charge of CPDNs and PDNs were measured with the Zetasizer Nano ZS (Malvern, UK) at room temperature. CPDNs and PDNs were newly synthesized; the suspension was added to a 200-mesh, carbon-plated copper grid and dried at room temperature. Then, we observed the copper grid with TEM (Tecnai G2F20S-TWIN, FEI, OR, USA). The UV-Vis spectrum was obtained with a UV-Vis spectrophotometer (Lambda 750, PerkinElmer, MA, USA). We used a FL spectrometer (LS55, PerkinElmer, MA, USA) to obtain PL spectra. A thermal infrared imager (Ti27, Fluke, WA, USA) was used to irradiate 300 μg/mL CPDNs with a 808 nm laser (0.8 W/cm^2^) to determine the in vitro photothermal effect of CPDNs.

### 4.6. Determination of DTX Encapsulation Rate in CPDNs

The encapsulation rate of DTX in CPDNs was determined by referring to a previously reported method [43]. CPDNs were dissolved by adding acetone followed by an intense vortex for 60 s. After sonication for 30 min, the mixture was centrifuged for 20 min at a speed of 2000 g/min to separate the supernatant. Then, the precipitate was added with acetone, and the same procedure was repeated. The supernatants collected by two centrifugation steps were evaporated. The polymer was removed with the addition of methanol to the system, vortex for 60 s, and centrifugation for 20 min at a speed of 2000 g/min. Docetaxel was then quantified in the supernatant (*W*) using the high-performance liquid chromatography (HPLC) method. The initial weight of DTX added to prepare CPDXs was termed as *W*_0_. The encapsulation rate of DTX was calculated as follows:$$\mathrm{Encapsulation}\;\mathrm{rate}\;\left(\%\right)=\frac{W}{W_{0}}\times 100\%$$

### 4.7. Release Percentage of DTX in CPDNs

To estimate the cumulative release of DTX in CPDNs, CPDNs were added in sealed dialysis membranes (8–14 KDa) and incubated in PBS with 0.5% polysorbate at 37 °C. Samples were taken at the point of 1 h, 2 h, 4 h, 8 h, 24 h, 48 h, and 72 h, respectively, to estimate the weight of DTX (*Wr*) by the HPLC analysis method. The initial weight of DTX in the dialysis bag was termed as *W*_1_. The release percentage of DTX in CPDNs was estimated as follows:$$\mathrm{Cumulative}\;\mathrm{release}\;\left(\%\right)=\frac{W_{r}}{W_{1}}\times 100\%$$

### 4.8. Protein Identification of CPDNs

We analyzed the western blot of the protein profile of DU145 cells and CPDNs. In brief, an NP-40 lysis buffer, phenylmethylsulfonyl fluoride (PMSF), and protease inhibitor were added into cells to totally disrupt cells on ice. Then, the mixture was centrifuged for 10 min at 12 × 10^3^ rpm/min to obtain the total protein in the supernatant. The protein concentration was determined by the BCA Protein Assay Kit. The protein was dissolved on a 12.3% sodium lauryl sulfate-polyacrylamide gel and then electrophoresed and transferred to a polyvinylidene fluoride membrane. It was then block blotted with 5% skim milk at room temperature for 2 h and incubated with specific antibodies overnight. After washing the membrane, it was incubated with a goat antirabbit secondary antibody for 1 h at room temperature to enhance the chemiluminescence detection of specific protein bands.

### 4.9. Intracellular Localization of PDNs and CPDNs

PDNs were labeled with FITC, 2 mg PDNs were precipitated in 1 mL deionized water, and 0.1 mL NH_3_·H_2_O and 4 mg APTES were added, left standing for 24 h and centrifuged and washed 3 times. Then, the NH2-PDNs were incubated with 0.1 mg/mL FITC, in the dark for 72 h, and rinsed 3 times. An amount of 1 mL 0.1 mg/mL FITC-PDNs and 1 mL 0.1 mg/mL FITC-CPDNs were incubated with DU145 cells, and flow cytometry and CLSM were detected for 1, 3, 5, 12, and 24 h.

### 4.10. Stability Test

To observe the stability, the CPDNs were incubated with PBS and RPMI-1640 supplemented with 10% fetal bovine serum (FBS) at 37 °C with a concentration of 20 μg/mL; we recorded the size variation at the points of 0.5 d, 1 d, 2 d, 3 d, 4 d, 5 d, 6 d, and 7 d.

### 4.11. Qualitative Observation of In Vitro Cellular Uptake

PDNs and CPDNs are labeled with ICG. Briefly, PLGA and an appropriate amount of drug were dissolved in ethyl acetate to obtain a solution of 10% (*w/v*) PLGA and 1.5 mg/mL (*w/v*) docetaxel. Then, 150 μL ICG solution (3 mg/mL) was added to the mixture, which was dissolved in a 2.2% (*w/v*) polyvinyl alcohol (PVA) solution. Then, the mixture was vigorously shaken and ultrasonicated. The resulting nanosuspension was magnetically stirred overnight to allow the organic solvent to evaporate. Continuous ultracentrifugation/washing with distilled water can obtain the purification of ICG-labelled PDNs. Finally, the ICG-PDNs were encapsulated with a cancer cell membrane to prepare ICG-CPDNs according to the CPDN synthesis method described above. Using labeled PDNs and CPDNs to prepare CPDNs, we qualitatively observed the uptake of nanoparticles by DU145 and PrEC cells. Next, an appropriate amount of cells were inoculated in a 12-well cell culture plate and cultured for 24 h. Then, we removed the old medium and added 1 mL of serum-free, fresh medium containing PDNs or CPDNs (PLGA concentration is 0.2 mg/mL). All samples were observed under a laser confocal scanning microscope.

### 4.12. Quantitative Detection of In Vitro Cellular Uptake

PDNs were labeled with ICG, and CPDNs were prepared with labeled PTNs. We quantitatively detected the uptake of nanoparticles by DU145 and PrEC cells. Using a 12-well cell culture plate, the cell culture wells were divided into a PDN group, a CPDN group, and a blank group; the holes were repeated for 3 holes. We inoculated 1 mL of cell liquid with a cell density of 2 × 10^5^ cells/mL per well. After 24 h, 1 mL of serum-free medium containing PDNs or CPDNs (PLGA concentration of 0.2 mg/mL) was added to the PDN group and CPDN group, respectively. In the blank group, 1 mL of serum-free medium without drugs was added. After incubating for 2 h, each group was trypsinized and washed 3 times with PBS. Finally, it was resuspended in 500 µL PBS, and the cell uptake was detected by FL-2 channel flow cytometer.

### 4.13. Cytotoxicity Test

DU145 cells were incubated with 1 mg/mL (0, 20, 40, 60, 80 µL) PDNs and CPDNs for 24 h. The supernatant was removed, and each chamber was rinsed with PBS (pH = 7.4) for 3 times. Then, MTT (100 µL, 5 mg/mL) in PBS was added to each well and incubated with the cells at 37 °C for 4 h. After removing the supernatant, DMSO (200 µL/well) was added to dissolve the succinate dehydrogenase in living cells. Samples were measured with a 1420 multilabel counter at a wavelength of 490 nm.

### 4.14. Animal Model and Tumor Formation Assay

BALB/C nu/nu male mice (6 weeks) were purchased from Shanghai SLAC laboratory animals. All animal experiments were carried out in accordance with the “Guidelines for the Care and Use of Laboratory Animals” of the National Institute of Health and were approved by the Ethics Committee of Zhongda Hospital Affiliated to Southeast University. The cells were collected, resuspended at 2 × 10^7^ cells/mL with Matrigel (BD), prepared with 0.1 mL of cells (2 × 10^6^ cells in total) through skin alcohol, and inoculated subcutaneously on both sides of nude mice. An intravenous injection of normal saline 200 µg/mL, docetaxel, PDN, or CPDN dispersion 200 µL was used. We calculated the tumor volume by (length × width^2^)/2 every week. After 4 weeks, all tumors were collected to measure their volume. The tumor was immediately fixed with formalin and then removed and embedded in paraffin.

### 4.15. In Vivo Tumor Imaging

There were 5 mice in each group. The tumor can be imaged 7 days after the establishment of the DU145 tumor-bearing nude mouse model, which can be verified by injection of fluorescein in the IVIS spectral imaging system (Caliper Life Sciences, Hopkinton, MA, USA). Then, 200 µL of normal saline, docetaxel, PDN, or CPDN dispersion was injected intravenously, and small animal live imaging was performed under the same conditions at 0, 1, 2, 4, 8, and 24 h after injection. The mice were sacrificed 24 h after the injection of nanoparticles. We collected the main organs (liver, kidney, and tumor) and used the IVIS spectrum detection of the in vivo fluorescence imaging system (Caliper Perkin Elmer).

### 4.16. SPECT-CT

^125^I labeling of PDNs/CPDNs was performed with ^125^I labeling by the classical iodine-catalyzed method 22: add 1 mg PDNs/CPDNs to 50 µL of PBS (0.01 m, pH = 7.4) and add to a glass tube coated with 20 µg of iodine. Then, we added the freshly prepared ^125^I (Na^125^I solution) into the tube, and shook the tube intermittently to avoid precipitation of PDNs/CPDNs. This reacted at room temperature for 10 min. Chromatographic paper and 0.9% NaCl were used as the stationary phase and mobile phase, respectively. The in vitro stability of the PDNs/CPDNs in the 0.1% fetal calf serum solution was tested at 37 °C for 24 h. Finally, the labeling ability of ^125^I was determined according to the requirement of labeling rate > 90%. In order to determine the biodistribution and metabolism of PDNs/CPDNs, 1-µL embolization bead suspension (containing 100 µg Au and 3.7 MBq ^125^I) was directly injected into the center of the tumor tissue with a 1-µL syringe. SPECT/CT was used to monitor the biodistribution of PDNs/CPDNs. The scanning parameters were as follows: SPECT: collection count 5 × 10^5^, matrix 256 × 256, magnification 2; CT: voltage is 130 kv, current is collected after automatic exposure control, and the pitch is 1. The mice were anesthetized with 50–75 µL pentobarbital (3%) and scanned at 0, 1, 2.5, 8, 18, 24, 48, and 90 h pi. The mice remained awake during the two scans. We recorded the injection point and thyroid count based on the plane image. At the above time point, the residual radioactivity of the whole mouse was measured with a radioactivity calibrator. An amount of 1 µL of Na125I solution (3.7 MBq) was injected into the center of the mouse tumor tissue as a reference.

### 4.17. Immunohistochemical Staining (IHC)

According to the manufacturer’s instructions, the paraffin-clad xenograft tumors were stained with anti-Ki-67 and CD34 (both 1:100, Abcam) antibodies and then observed under a microscope.

### 4.18. Statistical Analysis

All data are reported as mean ± standard deviation (sd). Statistical analysis was performed using SPSS 20.0 software (New York, NY, USA). The Student’s *t* test was used to compare the two groups. One-way analysis of variance (ANOVA) and Tukey’s post hoc test were used to compare multiple groups.

## Figures and Tables

**Figure 1 ijms-23-03623-f001:**
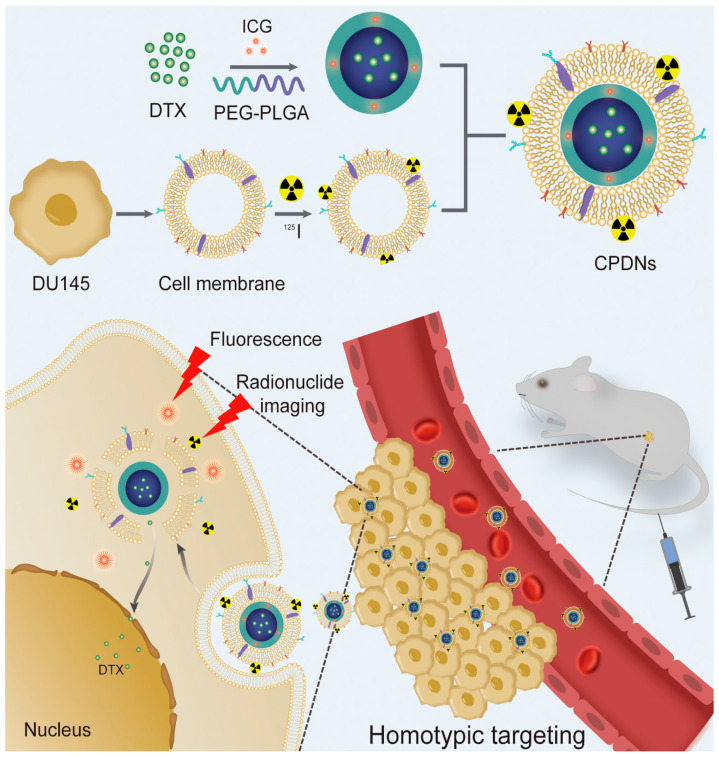
Schematic representation of CPDNs’ fabrication.

**Figure 2 ijms-23-03623-f002:**
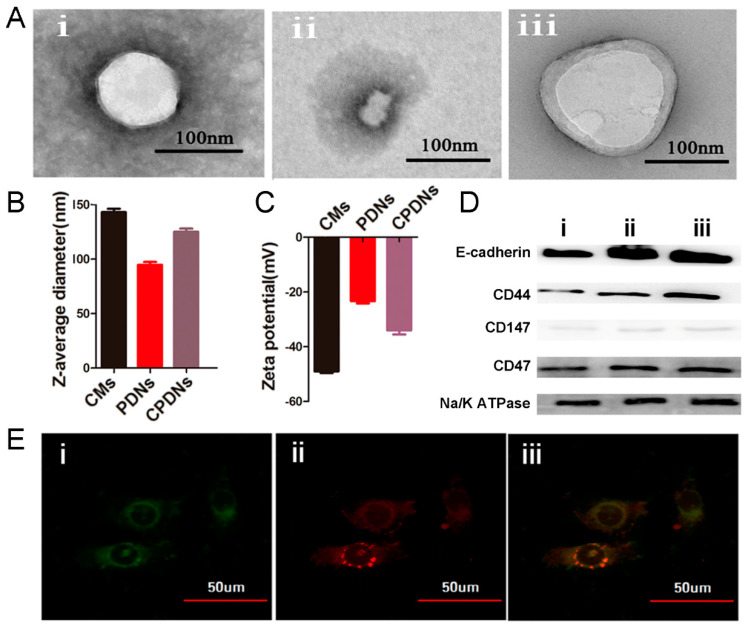
Characterization of the CPDNs: (**A**) the transmission electron microscopic images (TEM) of (**i**) PDNs, (**ii**) DU145 cell membrane, and (**iii**) CPDNs. Samples were negatively stained with uranyl acetate. All scale bars = 100 nm. (**B**,**C**) Particle size and zeta potential of the CMs, PDNs, and CPDNs. (**D**) Western blotting analysis for DU145 cells’ membrane-specific protein markers: (**i**) DU145 cells, (**ii**) DU145 cell membrane, and (**iii**) CPDNs. (**E**) Colocalization of PDNs and CMs upon cellular uptake. CPDNs were fabricated with PDNs loaded with DiD (red channel), and the membrane was labeled with FITC (green channel). All channels were deconvolved by software to eliminate out-of-focus fluorescent signal. The yellow color represents colocalization of PDNS and CMs signals. Data are given as mean ± SD (*n* = 3).

**Figure 3 ijms-23-03623-f003:**
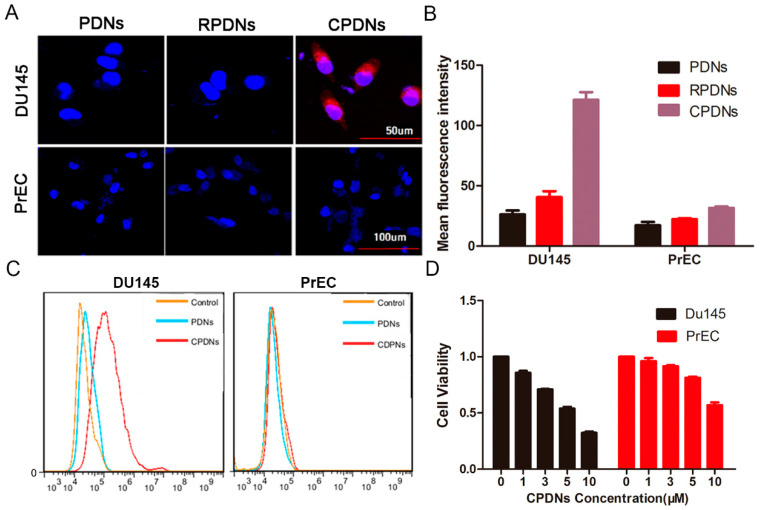
(**A**) Images of cellular uptake of PDNs, RPDNs, CPDNs in DU145 cells and PrEC cells; (**B**) quantitative analysis after 1 h incubation. The nucleus was stained with Hoechst 33,342 (blue). The CPDNs were labeled with ICG. (**C**) Flow cytometry analysis of the internalization of PDNs (blue) and CPDNs (red) by DU145 cells and PrEC cells. (**D**) The MTT assay of DU145 cells and PrEC cells incubated with different concentrations of CPDNs.

**Figure 4 ijms-23-03623-f004:**
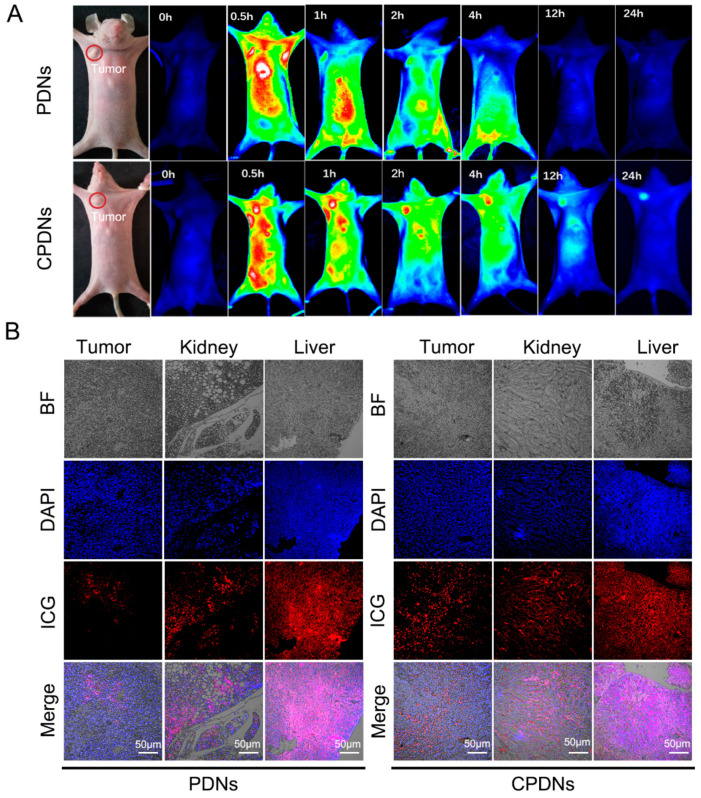
Biocompatibility effect in vivo: (**A**) in vivo distribution of PDNs and CPDNs in DU145-implanted tumor mice model examined by fluorescence imaging at different time points; (**B**) immunofluorescence of tumor, kidney, and liver slices 24 h post injection with PDNs and CPDNs groups.

**Figure 5 ijms-23-03623-f005:**
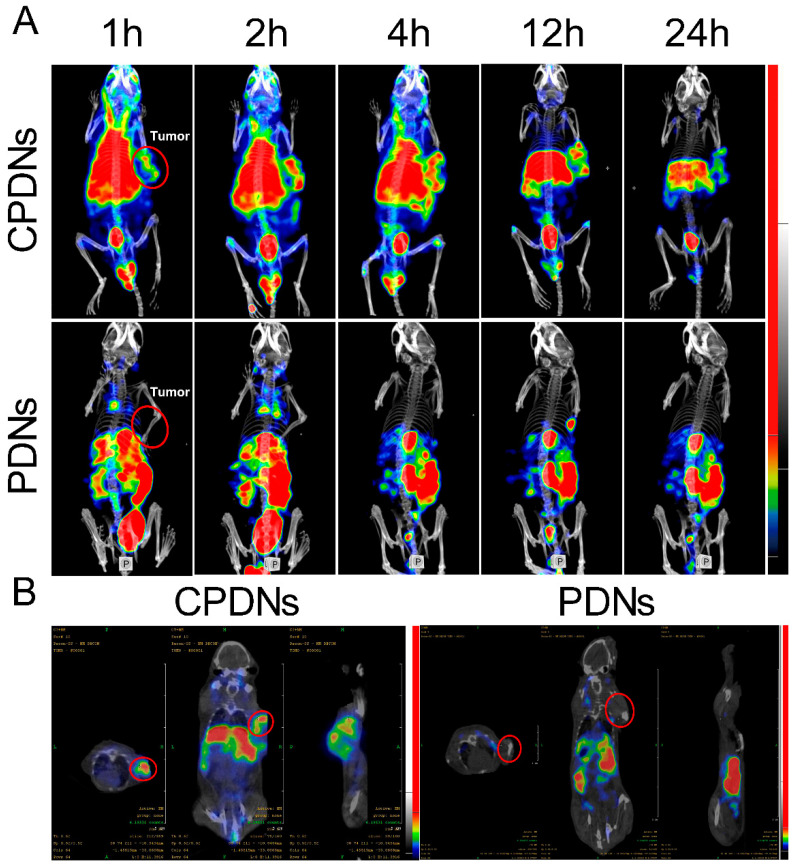
Biocompatibility effect in vivo: (**A**,**B**) the imaging of SPECT-CT of bioaccumulation of PDNs and CPDNs in implanted tumor mice postinjection.

**Figure 6 ijms-23-03623-f006:**
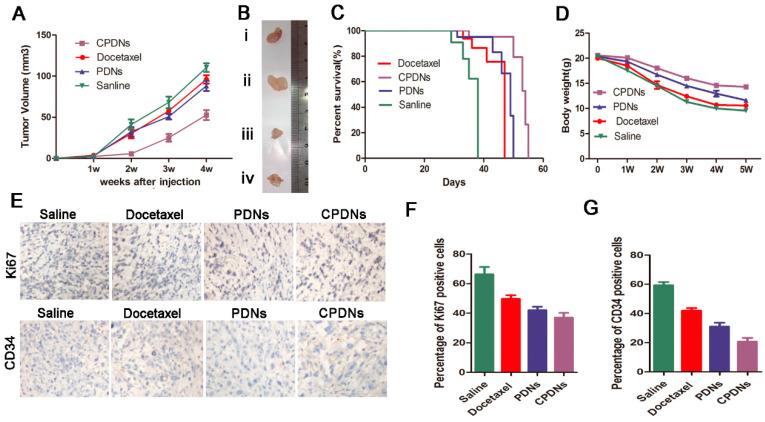
Treatment of CRPC: (**A**) relative tumor growth curves (**B**) and representative images after various treatments (from top to bottom, (**i**) Docetaxel; (**ii**) saline; (**iii**) CPDNs; (**iv**) PDNs). (**C**,**D**) Mice median survival time and body weight changes after different treatments. (**E**–**G**) CD34 and Ki-67 immunostaining of the tumor tissues at the end of the experiment, and the CPDNs group reduced CD34 and Ki-67 immunostaining in tumors.

## Data Availability

The data presented in this study are available on request from the corresponding author.

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
