# Peer review of "CRPC Membrane-Camouflaged, Biomimetic Nanosystem for Overcoming Castration-Resistant Prostate Cancer by Cellular Vehicle-Aided Tumor Targeting"

_ijms, 2022, doi:10.3390/ijms23073623_

Round 1

Reviewer 1 Report

The manuscript by Lu et al. describes the development of an original biomimetic nanocomposite with a PEG-PLGA core containing docetaxel covered by cell membrane camouflage.
The aim of the project and the results obtained are significant.

Some inaccuracy in the manuscript have to be addressed before publication.

  • fig S1 and fig S3 are inverted
  • line 157: what is ICG? in the fig. 3 caption and in the experimental section it is indicated that the NPs were labelled with nile red
  • line 159: specify here the acronymous PrEC and, more generally, reformulate the results displayed in fig. 3
  • please maintain the same nomenclature for the different NPs along the entire manuscript (i.e. in section 2.2 red blood cell membrane coated PDNs are called RBCPDNs but in the relative fig. 3 they are called RPDNs)

Author Response

  1. fig S1 and fig S3 are inverted

Response: Thank you for pointing out the mistake, we have adjusted the order of S1 and S3, and S3 were adjusted as S5 in revised Supplementary Information.

  1. line 157: what is ICG? in the fig. 3 caption and in the experimental section it is indicated that the NPs were labelled with nile red

Response: Thank you for your critical comment. We apologize for the wrong description of the dye encapsulated in NPs. ICG (Indocyanine green) is a kind of near infrared fluorescence dye, which has been proven to be a relatively safe drug. Actually, what we used in fig. 3 to label CPDNs and PDNs was ICG, and we have carefully checked and corrected all related description in the caption of fig.3 and the experimental section (Page 6, line 180-181; Page 11, line 350-352 and Page 12, line 353-358) with yellow background. 

  1. line 159: specify here the acronymous PrEC and, more generally, reformulate the results displayed in fig. 3

Response: Thank you for your professional comment. The acronymous PrEC in line 159 was a normal human prostate cell line, which was used as a negative control in our work. We have added this specified description in revised manuscript (Page 5, line 163-164) and reformulated the results displayed in fig.3 (Page 5, line 167-168 and Page 6, line 169-170) with yellow background.  

  1. please maintain the same nomenclature for the different NPs along the entire manuscript (i.e. in section 2.2 red blood cell membrane coated PDNs are called RBCPDNs but in the relative fig. 3 they are called RPDNs)

Response: Thank you for your kind suggestion. We apologize for not noticing this inconsistent nomenclature. We have amended all RBCPDNs as RPDNs in revised manuscript.

Reviewer 2 Report

The work by Lu et al. is of high importance for the development of nanodrugs.

It is well organized and the whole experimental set-up clearly provides the concept.

My only question is whether there is a "recipe" or some general guidelines in order to develop nanocomposites with tailored biocompatibility.For example, which are the criteria and/or parameters for the development of such materials? This would further promote the strength of the article.

Author Response

I am very grateful to your kind comments for the manuscript.The cancer-cell-biomimetic nanoparticles in this study could serve as a smart nanoplatform for the treatment of homotypic metastatic cancers,however,Such study was still in the exploratory stage and lacked unified quality control. We tried our best to describe our experimental conditions and methods in detail in this article.

Reviewer 3 Report

This work is devoted to the study of a biomimetic nanosystem with a camouflage membrane to overcome castration-resistant prostate cancer by targeting the tumor using a cell carrier. The article is well written and easy to read. There are some points that you should pay attention to:
1. In the introduction, please add more comparison with similar systems.
2. Figure 2 is very large. It is desirable to divide it into parts and at the same time discuss each part in detail with references to the literature.
3. Many experimental data need a more detailed explanation. Please use literature references in discussing your results.
4. Please make your conclusions more concise.

In general, the article is interesting and worthy of acceptance after minor revision.

Author Response

  1. In the introduction, please add more comparison with similar systems.

Response: Thank you for your professional comment. We have added more comparison analysis between blood cell membranes and CRPC cell (DU145) membranes used in our work and corresponding literature reference [32-33] (Nano Today 2020, 35, 100986; Acs Nano 2019, 13 (5), 5591-5601), which was shown in revised manuscript with yellow background (Page 2, line 69-78).

  1. Figure 2 is very large. It is desirable to divide it into parts and at the same time discuss each part in detail with references to the literature.

Response: Thank you for your constructive comment. We have divided Figure 2 into three parts, respectively Figure 2, Figure S2 and Figure S3. Meanwhile, we added literature [34-35] (Adv. Mater 2018, 30 (23), 1706759; Acs Nano 2018, 12 (8), 8520-8530) in discussion part.

  1. Many experimental data need a more detailed explanation. Please use literature references in discussing your results.

Response: Thank you for your professional comment. According to experimental data, we have added literature reference in experimental data explanation [34-37] in revised manuscript. In the cumulative release experiment (fig. S2), we proved the release percentage of DTX from CPDNs was lower than that from PDNs in both PBS and 1640 cell culture medium, which might be due to the outer shell membrane acting as a diffusion barrier for drug diffusion to release. In revised manuscript, we added literature references ( Adv. Mater 2018, 30 (23), 1706759; Acs Nano 2018, 12 (8), 8520-8530) in this explanation part to prove the protective effect of the outer membrane. In addition, in the evaluation experiment of treatment of CRPC, the relative tumor growth curves (fig. 6a) showed the antitumor growth of the PDN group was stronger than that of the DTX group, which was consistent with previous literature reports. To support this point, we added two literature reference ( J Control Release 2015, 220, 545-555; J. Drug Target. 2011, 19 (7), 516-527) to explain the better efficacy of PDN compared with DTX. 

  1. Please make your conclusions more concise.

Response: Thank you for your constructive comment. We have refined the conclusion part in revised manuscript (Page 9, line 240-256) with yellow background.

Reviewer 4 Report

The authors developed cell membrane-camouflaged polymeric nanoparticles for docetaxel delivery for prostate cancer therapy.  The theme is interesting and timely, since prostate cancer is one of the most prevalent type of cancers in the world. Cell membrane-camouflaged NPs are advantageous to prolong blood-circulation time, avoid the recognition by the immune system and to increase the tumor targeting ability. The developed NPs were physiochemically characterized using adequate methodologies, and their in vivo anticancer activity was studied in mice. However, several experiments details are missing, and some points should be addressed to improve the quality of the manuscript. I recommend major revisions before publication. Below the authors can find some suggestions and questions.

Keywords: please avoid repeating words from the title, such prostate cancer

Line 124: The authors mentioned that DTX encapsulation rate was 52.5%. Experimental details for its quantification are missing in the methods section.

Figure 2D: The authors present the release results, but there is no information on the experimental methodology for these experiments. The authors only provided the information about stability in these buffers (lines 315-320)? Did the release was performed in dialysis membranes or aliquots? Please add experimental details. Also, release experiments should be performed at physiological temperature (37 ºC) and not at 4ºC.

Figure 2F: the results of stability in FBS are missing (the authors mentioned that this was performed in lines 315-320). Also, in the methods the authors mentioned that the used cell culture medium for these experiments was DMEM and not RPMI-1640. Please clarify.

Lines 154-156: the authors stated that cell uptake of CPDNs was higher than for RBCPDNs. However, figure 3b reveals a much higher fluorescence for RBCPDNs. Please clarify.

Lines 316-317: Why did the authors study the stability of the NPs in PBS, FBS and DMEM at 4º? Stability at simulated physiological conditions should be observed at 37 ºC.

Lines 321-328: please include the information about the experimental procedure for protein extraction/isolation

Lines 336-339: information about stability experiments should be all in the same section. I suggest moving the text from lines 315-320 to this section.

Line 341: Please include the information about the experimental procedure for PDNs and CPDNs labeling with Nile Red and DiO, respectively.

Lines 359-365: Please change the verbs to the past tense.

Lines 395-405: Tumor xenograft model establishment should appear before the tumor imaging studies (366-374)

Author Response

  1. Keywords: please avoid repeating words from the title, such prostate cancer

Response: Thank you for your kind suggestion. We have replaced prostate cancer with oncotherapy in Keywords section.

  1. Line 124: The authors mentioned that DTX encapsulation rate was 52.5%. Experimental details for its quantification are missing in the methods section.

Response: Thank you for your kind comment. We apologize for missing details of the determination of DTX encapsulation rate. We have added this part in experimental details with yellow background in revised manuscript (Page 10, line 306; Page 11, line 307-317).

  1. Figure 2D: The authors present the release results, but there is no information on the experimental methodology for these experiments. The authors only provided the information about stability in these buffers (lines 315-320)? Did the release was performed in dialysis membranes or aliquots? Please add experimental details. Also, release experiments should be performed at physiological temperature (37 ºC) and not at 4ºC.

Response: Thank you for your professional comment and kind suggestions. We apologize again for not describing the information about the release of DTX. We performed the release analysis in dialysis membranes and detailed information was added in experimental details with yellow background in revised manuscript (Page 11, line 319-325). And the release experiment was performed at 37 ºC. Thank you again for your professional comments.

  1. Figure 2F: the results of stability in FBS are missing (the authors mentioned that this was performed in lines 315-320). Also, in the methods the authors mentioned that the used cell culture medium for these experiments was DMEM and not RPMI-1640. Please clarify.

Response: Thank you for your professional comment. In this work, DU145 cells was cultured with RPMI-1640 medium and the stability analysis was performed in PBS and RPMI-1640 medium with 10% FBS. In figure 2F, 1640 M referred to RPMI-1640 supplemented with 10% FBS. We have corrected the mistake in the description about stability analysis with yellow background in experimental section (Page 11, lines 346-348).  

  1. Lines 154-156: the authors stated that cell uptake of CPDNs was higher than for RBCPDNs. However, figure 3b reveals a much higher fluorescence for RBCPDNs. Please clarify.

Response: Thank you for pointing out this problem. We are sorry to confuse the legend between CPDNs and RPDNs in figure 3b and we have amended this mistake in figure 3b in the revised manuscript.

  1. Lines 316-317: Why did the authors study the stability of the NPs in PBS, FBS and DMEM at 4º? Stability at simulated physiological conditions should be observed at 37 ºC.

Response: Thank you for pointing out this question. The stability analysis of NPs was performed in PBS and RPMI-1640 supplemented with 10% FBS at 37 ℃ in this work. We sincerely apologize for the mistake about the stability method and we have amended this part in Page 11, line 346-348 with yellow background in the revised manuscript.

  1. Lines 321-328: please include the information about the experimental procedure for protein extraction/isolation

Response: Thank you for your kind suggestion. We have added the detailed information about the experimental procedure for protein extraction in revised manuscript (Page 12, line 328-332).

  1. Lines 336-339: information about stability experiments should be all in the same section. I suggest moving the text from lines 315-320 to this section.

Response: Thank you for your kind suggestion. We have gathered all information about stability experiment in Page 11, lines 346-348 with yellow background in revised manuscript.

  1. Line 341: Please include the information about the experimental procedure for PDNs and CPDNs labeling with Nile Red and DiO, respectively.

Response: Thank you for your critical comment. We apologize for the wrong description about the dye labelled in PDNs and CPDNs. In this work, PDNs and CPDNs were both labelled with near infrared fluorescence dye ICG in the observation of in vitro cellular uptake and in vivo tumor imaging. ICG was dissolved in the PLGA dispersion system, following the same protocol as the preparation of CPDNs. The detailed information about the experimental procedure for PDNs and CPDNs labeling with ICG was added in revised manuscript (Page 11, line 350-352; Page 12, line 353-358).

  1. Lines 359-365: Please change the verbs to the past tense.

Response: Thank you for your critical comment. We have changed the verbs to the past tense in revised manuscript (Page 12, line 376-381).

  1. Lines 395-405: Tumor xenograft model establishment should appear before the tumor imaging studies (366-374)

Response: Thank you for your critical comment. The order of information about tumor xenograft model establishment and tumor imaging studies have been readjusted.

Round 2

Reviewer 4 Report

The authors addressed all my comments. Publication is deserved.